# SIR-EN—New Biomarker for Identifying Patients at Risk of Endometrial Carcinoma in Abnormal Uterine Bleeding at Menopause

**DOI:** 10.3390/cancers16213567

**Published:** 2024-10-23

**Authors:** Carlo Ronsini, Irene Iavarone, Maria Giovanna Vastarella, Luigi Della Corte, Giada Andreoli, Giuseppe Bifulco, Luigi Cobellis, Pasquale De Franciscis

**Affiliations:** 1Department of Woman, Child and General and Specialized Surgery, University of Campania “Luigi Vanvitelli”, 80138 Naples, Italy; ireneiavarone2@gmail.com (I.I.); mariagiovanna.vastarella@unicampania.it (M.G.V.); andreoli.giada@gmail.com (G.A.); luigi.cobellis@unicampania.it (L.C.); pasquale.defranciscis@unicampania.it (P.D.F.); 2Department of Neuroscience, Reproductive Sciences and Dentistry, School of Medicine, University of Naples, 80138 Naples, Italy; luigi.dellacorte@unina.it (L.D.C.); giuseppe.bifulco@unina.it (G.B.)

**Keywords:** endometrial carcinoma, atypical hyperplasia, ultrasound, inflammation indices, diagnosis

## Abstract

This study investigated the relationship between systemic inflammatory reaction (SIR) indices and endometrial thickness in postmenopausal women with abnormal uterine bleeding. A new biomarker, SIR-En, combining SII (Systemic Inflammatory Index) and endometrial thickness, was developed. This study compared 192 patients with endometrial cancer and 50 with endometrial hyperplasia. The results showed that patients with cancer had significantly higher SIR-En values than those with hyperplasia (8710 vs. 6420; *p* = 0.003). The SIR-En index demonstrated moderate diagnostic ability for endometrial cancer, with an area under the ROC curve (AUC) of 0.6351 and a cut-off of 13,806, yielding high specificity (94%) and positive predictive value (96%). This suggests that SIR-En could aid in discriminating the endometrial carcinoma from atypical hyperplasia, improving diagnosis and treatment strategies.

## 1. Introduction

Abnormal uterine bleeding during menopause should be investigated for the possible risk of being affected by dysplastic or neoplastic disease [1]. In the presence of this symptom, an in-depth first-level examination represented by transvaginal pelvic ultrasound is necessary [2,3]. In this examination, the parameter of greatest interest is endometrial thickness. A cut-off of 4 mm is reported in the literature to discriminate patients deserving further diagnostic investigation [4,5]. The second-level investigation is hysteroscopy or curettage dilatation, which allows for the acquisition of a histological sample [6]. Endometrial lesions may be sustained by dysplastic conditions, such as endometrial hyperplasia, or neoplastic conditions, such as endometrial carcinoma [7]. The relationship between atypical hyperplasia and endometrial carcinoma is well-known and may represent its natural evolution [8,9]. On the other hand, it is also likely that such neoplastic progression may result in an immune response on the part of the patient [10]. Degeneration towards neoplastic forms can lead to alterations in the tumor microenvironment, plausibly linked to local infiltration, which can be registered as changes in the balance of the subject’s immune system [11]. From this point of view, a helpful biomarker is the systemic inflammatory reaction (SIR). These biomarkers have proven to be very useful from both a diagnostic and prognostic point of view for numerous solid tumors [12,13]. Combining systemic inflammation indices represented by SIR and the assessment of endometrial thickness can help identify patients at risk of neoplastic degeneration and optimize diagnostic timelines and therapeutic courses. For this reason, we conducted a retrospective multicenter study to evaluate the different distributions of a new biomarker based on the interaction between SIR and endometrial thickness in patients presenting with metrorrhagia, termed SIR-En.

## 2. Materials and Methods

### 2.1. Ethical or Institutional Review Board Approval

This study was conducted in two university clinics where all patients treated must sign a dedicated consent for anonymous data processing. The research methods were established a priori and authorized through evaluation by the Ethics Committee of the individual centers (IRB 30661/2022 of 31 March 2022)

### 2.2. Study Design

The research methods were established a priori. This study is a retrospective case–control analysis of women who suffered from abnormal uterine bleeding in menopause between August 2023 and January 2024, who were referred to AOU Vanvitelli and Policlinico Federico II in Naples, Italy. Patients’ data were reported from hospital medical records.

Inclusion criteria to be enrolled were menopausal condition, endometrial thickness ≥ 4 mm assessed by transvaginal ultrasound, with complete anatomopathological information obtainable, having received a complete formula peripheral blood sample within 7 days before definitive histological diagnosis, which has undergone hysterectomy for endometrial dysplasia or neoplasia. Exclusion criteria were patients with any chronic systemic inflammatory condition supported by any clinical picture, such as chronic inflammatory diseases (Chron’s disease, Rettocolitis, Lupus erythematosus, multiple sclerosis, Hashimoto’s thyroiditis, non-alcoholic hepatic steatosis, fibromyalgia, chronic renal insufficiency, hepatitis, osteoarthritis, and psoriasis); patients with an additional synchronous oncological diagnosis or within the previous 3 years; patients with corticosteroid overproduction disorders; and patients on steroid therapy within the last 30 days before blood sample. Patients with partial information or endometrial thickness < 4 mm and histological diagnosis of one of the two diseases of interest would have been considered as ‘Intention to treat’. No patients showed such characteristics.

### 2.3. Data Collection

Endometrial thickness was recorded by transvaginal pelvic ultrasound and expressed in millimeters. Peripheral blood samples were collected from the ulnar vein (3.0 mL) between 7 days before the surgery. These samples were placed in tubes containing ethylenediaminetetraacetic acid (EDTA). Hemoglobin (Hb), hematocrit (Hct), erythrocyte, total lymphocyte, monocyte, eosinophil, basophil counts, and platelet counts were measured and expressed as total count/Liter. The systemic inflammatory reaction (SIR) was evaluated through 4 different parameters: the neutrophil-to-lymphocyte ratio (NLR), which was calculated as the ratio of neutrophils to lymphocytes, the monocyte-to-lymphocyte ratio (MLR), and platelet-to-lymphocyte ratio (PLR), which were determined by dividing monocyte counts and platelet counts by lymphocyte counts, respectively. Moreover, the systemic inflammatory index (SII) was calculated by multiplying the neutrophil count by the platelet count and dividing by the lymphocyte count. Finally, a brand-new index, SIR-En, was calculated by multiplying SII and endometrial thickness.

### 2.4. Statistic Analysis

Using the Kolmogorov–Smirnoff test, the distribution of the continuous variables analyzed was checked.

The nominal variables were expressed as absolute frequency and percentages and compared using Fisher’s exact [14] and Chi-square tests [15]. Continuous variables were expressed as median and interquartile range and compared using the Wilcoxon test [16]. The variables were compared to more than two independent groups via Kruskal–Wallis [17].

Patients were divided according to histological diagnosis into hyperplasias and cancers.

The null hypothesis of our study was that there was no difference in the mean values of the SIR-En index between patients with endometrial hyperplasia and endometrialcCancer (H0: μ1 = μ2; H1: μ1 − μ2 ≠ 0 two sides). Secondary outcomes were the same evaluation for other SIR indexes and endometrial thickness.

We conducted a multivariate linear regression to demonstrate a correction between the parameters examined and the alterations in inflammation indices regression [18].

The significance of the model used was assessed using the maximum likelihood method [19].

We performed a ROC curve analysis to determine SIR-En’s diagnostic capability in predicting patient histology. We calculated the area under curve (AUC) for the ROC to assess the parameter’s overall diagnostic ability, with the 2000 bootstrap method [20]. The Youden Index was calculated to determine the optimal cutoff value for SIR-En [21].

The distribution of the continuous variables for the individual parameters of the reference outcome was graphed in boxplots. All statistical investigations were performed using R software and R Studio vers. 2023.12.1 + 402. ROC curves and AUCs were generated using the ROC package, and the Youden Index was calculated to determine the optimal cutoffs. The results were considered statistically significant at a *p*-value < 0.05. An anonymous dataset is reported in Appendix A.

## 3. Results

Between August 2023 and January 2024, 242 women were enrolled in this study. Based on anatomopathological data, patients were stratified into endometrial hyperplasia (50 patients) and endometrial cancer (192 patients). The two groups differed statistically significantly in age (59 vs. 62 years old, *p* < 0.001) and mean BMI (29 vs. 25, *p* = 0.008). The hyperplasia group consisted of 96% atypical hyperplasia. The carcinoma group had 90% endometrioid carcinomas. Moreover, 55% of the cancers already had myometrial infiltration at diagnosis, 6.7% had positive lymph nodes, and 18% of the carcinomas had microsatellite instability. The two groups showed a statistically significant difference in terms of mean neutrophils (4.15 vs. 4.98, respectively, for hyperplasia and cancers, *p* = 0.012) and endometrial thickness (10 vs. 14 mm, respectively, for hyperplasia and cancers, *p* = 0.001). The sample characteristics are summarized in Table 1.

### 3.1. Outcomes

The main outcome was the comparison of the mean of the SIR-En index in the two histological modalities.

Secondary outcomes were the distribution of NLR, MLR, PRL, and SII.

In the hyperplasia group, the SIR-En mean was statistically significantly lower compared to the cancer group (6420 vs. 8710; *p* = 0.003). All the other parameters failed to reach statistical significance, showing a worsening trend for SII (539 in the hyperplasia group vs. 622 in the cancer group; *p* = 0.2). Those results are summarized in Table 2.

The mean distribution of SIR-EN and its two founding parameters, SII and endometrial thickness, are graphed by boxplots in Figure 1.

### 3.2. Linear Regression

To assess the correlation between the histopathological nature and the indices of inflammation, we constructed a linear regression model including the three parameters (SIR-En, SIR, and endometrial thickness). The analysis showed a statistically significant relationship coefficient between the risk of cancer and endometrial thickness (Beta 3.9, CI 95% 1.5–6.3, *p* = 0.002), and with SIR-En (Beta 5547, CI 95% 1303–9791, *p* = 0.011). Those results are summarized in Table 3.

### 3.3. ROC Curve

To calculate the diagnostic capacity of the SIR-En index for endometrial carcinoma in abnormal uterine bleeding during menopause, we constructed a ROC curve, as shown in Figure 2.

The AUC was 0.6351 (95% CI: 0.5579–0.7121). Youden’s method was used to calculate the best diagnostic cut-off of SIR-En. The value was 13,805.83, rounded to 13,806.

This value showed a sensitivity of 0.349, a specificity of 0.940, a negative predictive value of 0.273, and a positive predictive value of 0.957. Table 4 summarizes the SIR-En cut-off’s diagnostic performance and relative confidence intervals.

## 4. Discussion

### 4.1. Interpretation of Results

Our study shows that patients with endometrial carcinoma, compared to patients with hyperplasia alone, show, on average, a higher SII and endometrial thickness. Combining these two parameters in the new SIR-En index optimizes their diagnostic performance. The patients in the endometrial carcinoma group showed statistically solidly higher mean SIR-En values. The linear regression showed that the correlation between indices of inflammation and endometrial thickness has reason to be summarized in this new index. The pathophysiological reasons for this correlation may be twofold. On the one hand, endometrial thickness is greater in cancer patients, probably due to the higher replicative rate of the neoplastic cells [5]. On the other hand, the phenomena of neoplastic degeneration and local invasion may represent the first trigger for the immune system, which translates into alterations in the indices of inflammation [13]. A good part of the sample from the cancer arm showed myometrial infiltration, positivity of the lymphovascular spaces, or lymph node positivity, testifying to an extension outside the tissue of origin by the cancer and, therefore, a necessary interaction with the immune system, and represent themselves risk factors [22,23,24,25].

### 4.2. Clinical Implication

The SIR-En index was constructed to work in patients with abnormal menopausal uterine bleeding and endometrial thickening. Having an index that can screen patients with organic endometrial lesions in the presence of abnormal uterine bleeding at menopause can indicate the optimization of diagnostic and therapeutic pathways. When faced with a patient presenting with this symptomatology and endometrial thickening, the ability to screen for endometrial carcinoma using a simple blood sample can improve clinical practice. The area under the ROC curve for SIR-En shows results that are not particularly satisfactory, given the value of 63%. However, the derived cut-off value of 13,806 shows an excellent performance due to its high specificity (94%) and high positive predictive value (96%). Combining these two values makes it a valuable aid in screening for endometrial cancer patients with this clinical presentation. Such screening can translate into more in-depth diagnostic investigations in patients at higher risk [26] and the optimization of healthcare resources, especially in peripheral centers, with the possibility of centralizing the most suspicious cases [27]. It should also be considered that most diagnoses of metropathy are made by endometrial biopsy [28]. This biopsy does not necessarily represent the entire pathological picture with a risk of up-staging [29,30]. In this scenario, the treatments chosen for dysplastic pathologies such as hyperplasia may lead to surgical under-staging, such as the absence of lymph node investigations, conditioning the treatment course [31,32,33]. Integration with this new index may help to minimize this occurrence. It should also be considered that almost the entire sample with hyperplasia was represented by its atypical variant and, therefore, was at greater risk of degeneration. This fact reinforces the scientific evidence from using SIR-En, testifying that the neoplastic transformation and subsequent progression determines values above the cut-off. Finally, it must be emphasized that this study focused on patients with metrorrhagia and ultrasounds that indicated possible endometrial neoformation. Still, we can plausibly hypothesize that this index may have a diagnostic function even in asymptomatic patients [34].

### 4.3. Strenght and Limitations

The strength of our study lies in the high statistical significance of the results obtained and the high positive predictive value and specificity of the cut-off. This makes its use reassuring in identifying true negative subjects and provides a strong detection of cancer in positive cases. Limitations, on the other hand, are represented by its retrospective nature that does not allow for the complete elimination of certain confounders that may be linked to alterations in the immune system and the inflammatory response; despite the strict exclusion criteria and linked territoriality being limited to a single geographical area, they could expose the sample to environmental factors that may interact with the immune system [35]. Finally, a limitation is represented by the patient setting. All the patients included in the study were retrospectively enrolled based on a proven metro at a post-hysterectomy histological examination. This implies that our index can only be used for an organic lesion because it was studied in this patient setting. This limitation will be overcome by constructing prospective studies that include patients with no dysplastic or neoplastic diagnosis and our study should be considered as a pilot study.

### 4.4. Comparison with Existing Literature

The indices of inflammation that we have examined may be linked with carcinogenesis and tumor progression, either directly or through a mediating effect of other clinical conditions that favor both the pro-inflammatory state and endometrial carcinoma. Obesity is a known risk factor for non-oestrogenic hyperestrogenism and consequentially for endometrial carcinoma, but it is also associated with a chronic inflammatory state [36]. Individuals with a genetic predisposition to endometrial carcinoma, such as individuals with Lynch syndrome, may also have an imbalance in the inflammatory response. Recent studies have shown that defects in mismatch repair (MMR) genes can lead to increased production of neoantigens that stimulate a more robust immune response [37]. Our study confirms that the inflammatory status of patients is a step involved in the carcinogenesis and progression of endometrial carcinoma and is clinically detectable.

## 5. Conclusions

Our study shows how endometrial thickness detected on transvaginal pelvic ultrasound and SRI can be combined to help discriminate patients with dysplasia from patients with carcinoma in cases of abnormal menopausal bleeding. Nevertheless, in light of the retrospective nature, further studies with prospective design will be necessary to validate this relationship and optimize the clinical contextualization of the data.

## Figures and Tables

**Figure 1 cancers-16-03567-f001:**
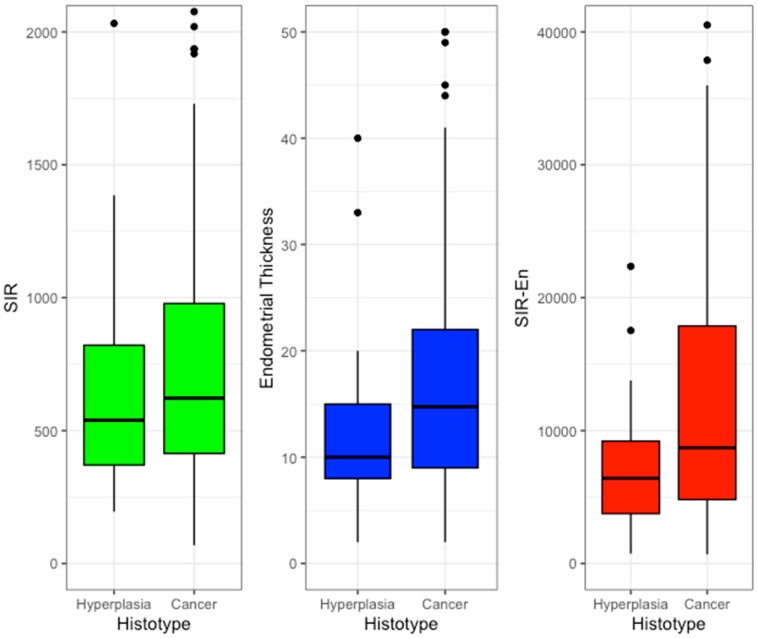
Boxplots.

**Figure 2 cancers-16-03567-f002:**
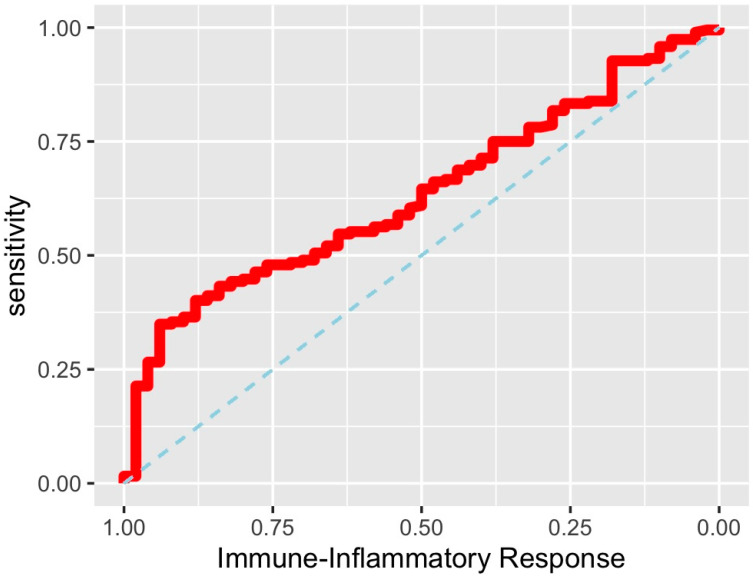
ROC curve for SIR-EN.

**Table 1 cancers-16-03567-t001:** Patient’s characteristics.

Characteristic	Hyperplasia, N = 50 ^1^	Cancer, N = 192 ^1^	*p*-Value ^2^
Age	59, (17)	62, (14)	**<0.001**
BMI	29, (9)	25, (11)	**0.008**
MSI			
MSI	-	31, (18%)	
MSS	-	138, (82%)	
Missing	-	23	
LVSI			
Negative	-	130, (70%)	
Positive	-	57, (30%)	
Missing	-	5	
Grading			
1	-	65, (34%)	
2	-	68, (36%)	
3	-	56, (30%)	
Missing	-	3	
Hystology			
Endometrioid	-	171, (90%)	
Other	-	7, (3.7%)	
Serous	-	11, (5.8%)	
Tipical Hyperplasia	2 (4%)	-	
Atypical Hyperplasia	48 (96%)	-	
Myometrial Infiltration			
No Infiltration	-	83, (45%)	
<50%	-	63, (34%)	
≥50%	-	38, (21%)	
Missing	-	8	
Lymphnodes			
Negative	-	167, (93%)	
Positive	-	12, (6.7%)	
Missing	-	13	
Neutrophils	4.15, (2.28)	4.98, (2.53)	**0.012**
Lymphocytis	1.80, (0.79)	1.97, (1.05)	0.4
Monocytis	0.50, (0.30)	0.49, (0.24)	0.3
Eosinophils	0.10, (0.10)	0.10, (0.12)	0.082
Platets	253, (86)	256, (92)	0.7
Endometrial Thickness	10, (7)	15, (13)	**0.001**

^1^ Median (IQR); n, (%); ^2^ Wilcoxon rank sum test; Fisher’s exact test. Bold was used for statistically significant value (*p* < 0.05).

**Table 2 cancers-16-03567-t002:** Outcomes.

Characteristic	Hyperplasia, N = 50 ^1^	Cancer, N = 192 ^1^	*p*-Value ^2^
SIR-En	6420, (5453)	8710, (13,049)	0.003
SII	539, (450)	622, (564)	0.2
NLR	2.12, (1.32)	2.33, (1.77)	0.2
MLR	0.22, (0.19)	0.22, (0.15)	>0.9
PLR	138, (77)	131, (67)	0.7

^1^ Median (IQR); ^2^ Wilcoxon rank sum test.

**Table 3 cancers-16-03567-t003:** Linear regression.

	SII			Endometrial Thickness			SIR-En		
Characteristic	Beta	95% CI ^1^	*p*-Value	Beta	95% CI ^1^	*p*-Value	Beta	95% CI ^1^	*p*-Value
Diagnosis									
Cancer	174	−40, 387	0.11	3.9	1.5, 6.3	0.002	5547	1303, 9791	**0.011**

^1^ CI = Confidence interval. Bold was used for statistically significant value (*p* < 0.05).

**Table 4 cancers-16-03567-t004:** Youden cut-off.

Cut-Off SIR-En ^1^	Estim.	Low.Lim(95%)	Up.Lim(95%)
Sensitivity	0.349	0.282	0.421
Specificity	0.940	0.835	0.987
Pos.Pred.Val.	0.957	0.880	0.991
Neg.Pred.Val	0.273	0.208	0.346
LR+	5.816	1.909	17.718
LR−	0.693	0.611	0.785
Odds ratio	8.397	2.518	28.000
Youden index	0.289	0.195	0.383
Accuracy	0.471	0.407	0.536
Error rate	0.529	0.464	0.593

^1^ 13,806, calculated by Youden Index.

## Data Availability

All research data can be provided by corresponding author upon explicit request.

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
