# Peer review of "SIR-EN—New Biomarker for Identifying Patients at Risk of Endometrial Carcinoma in Abnormal Uterine Bleeding at Menopause"

_cancers, 2024, doi:10.3390/cancers16213567_

Round 1
Reviewer 1 Report
Comments and Suggestions for Authors
The manuscript is devoted to the description of the new Biomarker SIR-EN, which allows to identify patients at risk of endometrial carcinoma in abnormal uterine bleeding at menopause. The manuscript is well written.
I have a few comments.
1.You use several parameters of the systemic inflammatory reaction. Their choice should be justified. Provide links to authoritative sources confirming the informativeness of these parameters.
2.Describe the exclusion criteria for the study in more detail. In addition to inflammatory Bowel Disease, rheumatological pathologies, autoimmunity diseases, were other diseases characterized by a pronounced inflammatory reaction excluded?
3.It is recommended to supplement the discussion with an analysis of other prognostic markers (clinical, biological, genetic) that can be used to identify patients at risk of endometrial carcinoma, especially in terms of comparison with the approach you propose.
Author Response
- You use several parameters of the systemic inflammatory reaction. Their choice should be justified. Provide links to authoritative sources confirming the informativeness of these parameters.
- Thank you for taking the time to read our manuscript and for this observation. Our study examines the classical changes in white blood cells during systemic inflammation and the ratio between them expressed as neutrophil-to-lymphocyte ratio (NLR), which was calculated as the ratio of neutrophils to lymphocytes; the monocyte-to-lymphocyte ratio (MLR) and platelet-to-lymphocyte ratio (PLR), which were determined by dividing monocyte counts and platelet counts by lymphocyte counts. These parameters are widely used to judge a patient's ‘pro-inflammatory’ status in both the oncological and clinical fields. (Please see: Forget, P., Khalifa, C., Defour, J. P., Latinne, D., Van Pel, M. C., & De Kock, M. (2017). What is the normal value of the neutrophil-to-lymphocyte ratio?. BMC research notes, 10(1), 12. https://doi.org/10.1186/s13104-016-2335-5 and Varım, C., Acar, B. A., Uyanık, M. S., Acar, T., Alagoz, N., Nalbant, A., Kaya, T., & Ergenc, H. (2016). Association between the neutrophil-to-lymphocyte ratio, a new marker of systemic inflammation, and restless legs syndrome. Singapore medical journal, 57(9), 514–516. https://doi.org/10.11622/smedj.2016154). Finally, the last parameter used is the systemic inflammatory index (SII), calculated by multiplying the neu-trophil count by the platelet count and dividing by the lymphocyte count, then using another reading of the same white blood cell values examined. These indices have also found use in the field of gynaecological oncology (Please see: Ahn, J. H., Lee, S. J., Yoon, J. H., Park, D. C., & Kim, S. I. (2022). Prognostic value of pretreatment systemic inflammatory markers in patients with stage I endometrial cancer. International journal of medical sciences, 19(14), 1989–1994. https://doi.org/10.7150/ijms.78182). We hope that these further indications may have clarified the reasons why we examined these parameters
- Describe the exclusion criteria for the study in more detail. In addition to inflammatory Bowel Disease, rheumatological pathologies, autoimmunity diseases, were other diseases characterized by a pronounced inflammatory reaction excluded?
- Thank you for pointing out this point for improvement. We have extended the description of the exclusion criteria in the methods section in this way:
- Lines 86-90 “Exclusion criteria were patients with any chronic systemic inflammatory condition supported by any clinical picture, such as Chronic inflammatory diseases (Chron's disease, Rettocolitis, Lupus erythematosus, multiple sclerosis, Hashimoto's thyroiditis, non-alcoholic hepatic steatosis, fibromyalgia, chronic renal insufficiency, hepatitis, osteoarthritis, psoriasis)”
You will find an updated version of the manuscript with these changes
- It is recommended to supplement the discussion with an analysis of other prognostic markers (clinical, biological, genetic) that can be used to identify patients at risk of endometrial carcinoma, especially in terms of comparison with the approach you propose.
3.Thank you for this suggestion. We have enriched the discussion by adding a paragraph entitled ‘Comparison with existing literature’ and changed the following:
– Linee 259-270: “The indices of inflammation that we have examined may be linked with carcinogenesis and tumour progression, either directly or through a mediating effect of other clinical conditions that favour both the pro-inflammatory state and endometrial carcinoma. Obesity is a known risk factor for non-oestrogenic hyperestrogenism and consequentially for endometrial carcinoma, but it is also associated with a chronic inflammatory state [36]. Individuals with a genetic predisposition to endometrial carcinoma, such as individuals with Lynch syndrome, may also have an imbalance in the inflammatory response. Recent studies have shown that defects in miss-match repair (MRR) genes can lead to increased production of neoantigens that stimulate a more robust immune response [37]. Our study confirms that the inflammatory status of patients is a step involved in the carcinogenesis and progression of endometrial carcinoma and is clinically detectable.”
36. Onstad, M. A., Schmandt, R. E., & Lu, K. H. (2016). Addressing the Role of Obesity in Endometrial Cancer Risk, Prevention, and Treatment. Journal of clinical oncology : official journal of the American Society of Clinical Oncology, 34(35), 4225–4230. https://doi.org/10.1200/JCO.2016.69.4638
37. Mlecnik, B., Bindea, G., Angell, H. K., Maby, P., Angelova, M., Tougeron, D., Church, S. E., Lafontaine, L., Fischer, M., Fredriksen, T., Sasso, M., Bilocq, A. M., Kirilovsky, A., Obenauf, A. C., Hamieh, M., Berger, A., Bruneval, P., Tuech, J. J., Sabourin, J. C., ... Galon, J. (2016). Integrative Analyses of Colorectal Cancer Show Immunoscore Is a Stronger Predictor of Patient Survival Than Microsatellite Instability. Science, 351(6274), aaf6925. https://doi.org/10.1126/science.aaf6925
You will find an updated version of the manuscript with these changes
Overall, thank you for your valuable comments. We believe they have helped to improve the quality of our hand-writing. We hope our answers were comprehensive and we are open to any future discussions to improve the presentation of our data and results.

Reviewer 2 Report
Comments and Suggestions for Authors
This is an interesting study. However, some alterations and/or comments are necessary in the manuscript.
1. Considering that there is no “screening” for endometrial carcinoma in the general population, the sentence of 20-21 lines (“This suggests that SIR-En 20 could aid in screening for endometrial carcinoma, improving diagnosis and treatment strategies”) should be changed.
2. It should be clarified why in your material there is no distinct group of women with endometrial thickness ≥4mm and normal endometrium which could be a “regular” finding in the general population.
Author Response
- Considering that there is no “screening” for endometrial carcinoma in the general population, the sentence of 20-21 lines (“This suggests that SIR-En 20 could aid in screening for endometrial carcinoma, improving diagnosis and treatment strategies”) should be changed.
- Thank you for taking the time to read our manuscript and for this observation. Abbiamo cambiato la frase come segue:
-Line 20-21: “This suggests that SIR-En could aid in discriminating the endometrial carcinoma from atypical hyperplasia, improving diagnosis and treatment strategies.”
You will find an updated version of the manuscript with these changes.
- It should be clarified why in your material there is no distinct group of women with endometrial thickness ≥4mm and normal endometrium which could be a “regular” finding in the general population.
- Thank you for pointing out this point for improvement. The presence of ultrasonographic endometrial thickening ≥4mm was a criterion for inclusion in the study. As this was a prospective study, patients were only enrolled if endometriopathy was suspected. Consequently, the study did not include patients with an unsuspected endometrial thickness. To minimize this bias, we reviewed our case series of ‘intention to treat’ patients based on histological diagnosis. In our sample of both atypical hyperplasias and carcinomas, no patient showed an ultrasonographic endometrial thickness of less than 4 mm. Therefore, however, this was a diagnostic criterion; no patient with one of the diseases under investigation was excluded based on endometrial thickness. While acknowledging the possible bias, it did not invalidate the results obtained. We also stated in matherials:
Line 98-106 : “Patients with partial information or endometrial thickness <4mm and histological diagnosis of one of the two diseases of interest would have been considered as ‘Intention to treat’. No patients showed such characteristics”
You will find an updated version of the manuscript with these changes
Overall, thank you for your valuable comments. We believe they have helped to improve the quality of our hand-writing. We hope our answers were comprehensive and we are open to any future discussions to improve the presentation of our data and results.
